# Preparation of ZnO Piezoelectric Thin-Film Material for Ultrasonic Transducers Applied in Bolt Stress Measurement

**Yuxia Zhang** [1,†]**, Yanghui Jiang** [2,†]**, Chi Ma** [1]**, Jun Zhang** [2,*]⬤ **and Bing Yang** [2,*]

1    CGN New Energy Holdings Co., Ltd., Hong Kong; zhangyuxia@cgnpc.com.cn (Y.Z.);
     liujie2022@whu.edu.cn (C.M.)
2    School of Power and Mechanical Engineering, Wuhan University, Wuhan 430072, China;
     2021282080073@whu.edu.cn
*    Correspondence: zhangjun2010@whu.edu.cn (J.Z.); toyangbing@whu.edu.cn (B.Y.)
†    These authors contributed equally to this work.

**Abstract:** The measurement of bolt preload by using ultrasound can be accurate, convenient, and can realize the real-time monitoring of the change in the residual axial stress of a bolt during use. In order to realize the ultrasonic measurement of bolt preload, the use of zinc oxide (ZnO) piezoelectric thin-film material as an ultrasonic transducer material to stimulate an ultrasonic signal on the bolt is a feasible solution. In this paper, we choose to use RF magnetron sputtering technology to prepare ZnO piezoelectric thin-film materials and study the effects of sputtering power and target substrate distance on the structure and ultrasonic properties of ZnO piezoelectric thin films during the preparation process, in order to lay the foundation for realizing the application of ZnO piezoelectric thin films in the field of bolt preload measurement. The experimental results show that too-large sputtering power or too-small target substrate distance will result in the particles having too much kinetic energy during sputtering and exhibiting a structure of multiorientation growth, which excites ultrasonic longitudinal–transverse waves. A sputtering power of 600 W, sputtering time of 4 h, and target substrate distance of 100 mm are ideal experimental parameters for a ZnO piezoelectric thin-film material to be excited by an ultrasonic longitudinal wave signal, and its ideal operating frequency is 41 MHZ. These research results of bolt stress detection demonstrate good application prospects.

**Keywords:** ZnO piezoelectric thin film; sputtering power; target substrate distance; ultrasonic signal

## 1. Introduction

Bolts, as a widely used fastener, have the advantages of high reliability, low cost, and easy handling [1–3]. Bolt preload within a specified range ensures the proper functioning of engineered structures [4]. Traditional preload testing methods include the torque method [5,6], nut rotation method, yield point method, and resistance strain gauge method [7,8]. However, due to the friction of a contact surface, the elastic–plastic strain of a bolt, and other reasons, there are deficiencies in the accuracy and practicality of the traditional methods to a certain extent; a convenient and accurate bolt preload measurement technology is urgently needed. Back in the 1970s, some scholars proposed the ultrasonic measurement of bolt preload technology. Through the use of ultrasound to measure bolt preload without disassembling the bolt, there is no additional force to affect the bolt's working condition [9,10]. The relationship between ultrasonic velocity and stress can be expressed by the following basic formula:

$$V_{l\sigma}{}^2 = V_l{}^2 - \sigma[(\lambda + 2\mu)(4\lambda + 10\mu + 4m)/\lambda + 1\mu + 2J]/[(3\lambda + 2\mu)(\lambda + 2\mu)] \tag{1}$$

$$V_{t\sigma}{}^2 = V_t{}^2 - \sigma[2J - 2\lambda(m + \lambda + 2\mu)/\mu]/[\rho_0(3\lambda + 2\mu)] \tag{2}$$

where $V_{l\sigma}$ and $V_{t\sigma}$ represent the propagation velocity of a longitudinal wave and transverse wave under a stress state, respectively; $V_l$ and $V_t$ represent the propagation velocity of a longitudinal wave and transverse wave without stress; $\sigma$ means stress; $\lambda$, $\mu$ represent the Raman constant; $m$, $J$ represent the third-order elastic constant; and $\rho_0$ indicates the material density. Therefore, the technique of ultrasonically measuring bolt preload is an ideal method to realize the real-time monitoring of preload in the work of bolts.

Currently, in order to excite ultrasonic waves on a bolt, the main methods are attaching a piezoelectric ceramic patch on the bolt or using a probe coupled to the bolt [7]. The patch method has the risk of the patch falling off, while the probe coupling method is more cumbersome; these inconveniences limit the application of ultrasonic measurements of bolt preload technology in practical engineering. Generally, piezoelectric sensors are made in two ways—one is using bulk piezoelectric substrates such as quartz, $LiNbO_3$, or $LiTaO_3$, and the other one is using piezoelectric thin films such as PZT, ZnO, or AlN [11–13]. Among them, PZT requires high voltage polarization to force the internal dipoles to align. Both ZnO and AlN are hexagonal wurtzite structures, and their piezoelectric properties depend entirely on their crystal orientation without poles. ZnO piezoelectric materials have a high piezoelectric coefficient and high electromechanical coupling coefficient $K^2$, can grow on a variety of substrates, have low epitaxial growth temperature requirements, and have a stoichiometry and texture which are easier to control. Therefore, ZnO coatings with a stronger piezoelectric effect and which are easier to obtain can be selected as the acoustoelectric conversion layer for excited ultrasound [14–16]. Using the magnetron sputtering method to sputter ZnO material to the bolt's surface, the formation of a ZnO piezoelectric film is stable and reliable and the film does not easy to fall off [17,18]; by applying a high-frequency voltage at both ends of the bolt, it can be excited ultrasonically, and then the preload measurement of the bolt can be realized under the working conditions [19]. At present, the relevant research on the application of ZnO piezoelectric films in ultrasonic measurement is relatively scarce; at the same time, the use of ultrasound in bolt preload measurement technology still has much room for improvement. ZnO piezoelectric films for bolt preload measurement have broad application prospects, the preparation process of the study will be an important part of the realization of ZnO piezoelectric film application [20]. The preparation of a stable, reliable, electromechanically coupled ZnO piezoelectric thin-film material that can excite a specific ultrasonic signal is of positive and important significance.

Among the ultrasonic stress measurement methods, ultrasonic longitudinal waves are more sensitive to changes in stress and are often used in engineering applications. The monowave method of bolt preload measurement using pure longitudinal waves is the more commonly used method [6,21,22], while the biwave method of combined longitudinal and transverse wave detection can reduce the calibration steps and simplify the operation [23–25]. In this paper, ZnO piezoelectric thin-film materials are prepared using RF magnetron sputtering under different process parameters. The structural characterization of the produced ZnO thin films is carried out to establish the connection between the sample preparation process parameters and the sample structure. At the same time, the ultrasonic properties of ZnO thin films prepared under different parameters are measured to investigate the ultrasonic properties of ZnO piezoelectric thin films with different structures and to establish the connection between the structure of ZnO thin films and the type of excited ultrasonic signals. In this paper, we explore the optimal preparation parameters of the ultrasonic transducer material ZnO to lay the foundation for realizing the application of ZnO piezoelectric films in the field of bolt preload measurement.

## 2. Experimental Section

### 2.1. Thin-Film Deposition

In this paper, ZnO piezoelectric thin-film materials were prepared by using the RF magnetron sputtering method under different process parameters, as shown in Table 1. The target material used in this paper was a ZnO ceramic target with a diameter of 150 mm,

a thickness of 8 mm, and a purity of 99.99%. The substrate materials selected in this paper were round monocrystalline silicon (100) and rectangular stainless steel. The main components of the equipment were a control system, a vacuum chamber, a sputtering source, etchers, a gas circuit system, and a circulating cooling system. A schematic diagram of the equipment is shown in Figure 1a. Before the start of the experiment, the substrate was ultrasonically cleaned in alcohol for 10 min, dried, and loaded onto the sample holder; the vacuum chamber was heated to 200 °C; the air pressure was pumped to $7 \times 10^{-3}$ Pa; argon gas (99.99% purity) was passed in; the bias and arc power supply were turned on; and plasma etching was carried out on the substrate under the conditions of $-150$ V, 50% duty cycle, 1.0 Pa air pressure, and 90 A current in order to remove impurities adhered to the surface of the substrate. After etching, a gas mixture of argon and oxygen (99.99% purity) with a ratio of 1:1 was introduced, and the air pressure was controlled to be 2.5 Pa. Then, the RF power supply was turned on to prepare the ZnO piezoelectric coating.

**Table 1.** Preparation parameters of the ZnO thin films with different sputtering powers and target substrate distances.

| Experimental Parameters | Conditions |
|---|---|
| Temperature/°C | 200 |
| Sputtering power/W | 100, 200, 400, ⬚600⬚, 800, 900 |
| | ↓ |
| Target substrate distance/mm | 60, 80, 100 |
| Sputtering pressure/Pa | 2.5 |
| Argon oxygen ratio | 1:1 |
| Sputtering time/h | 4 |

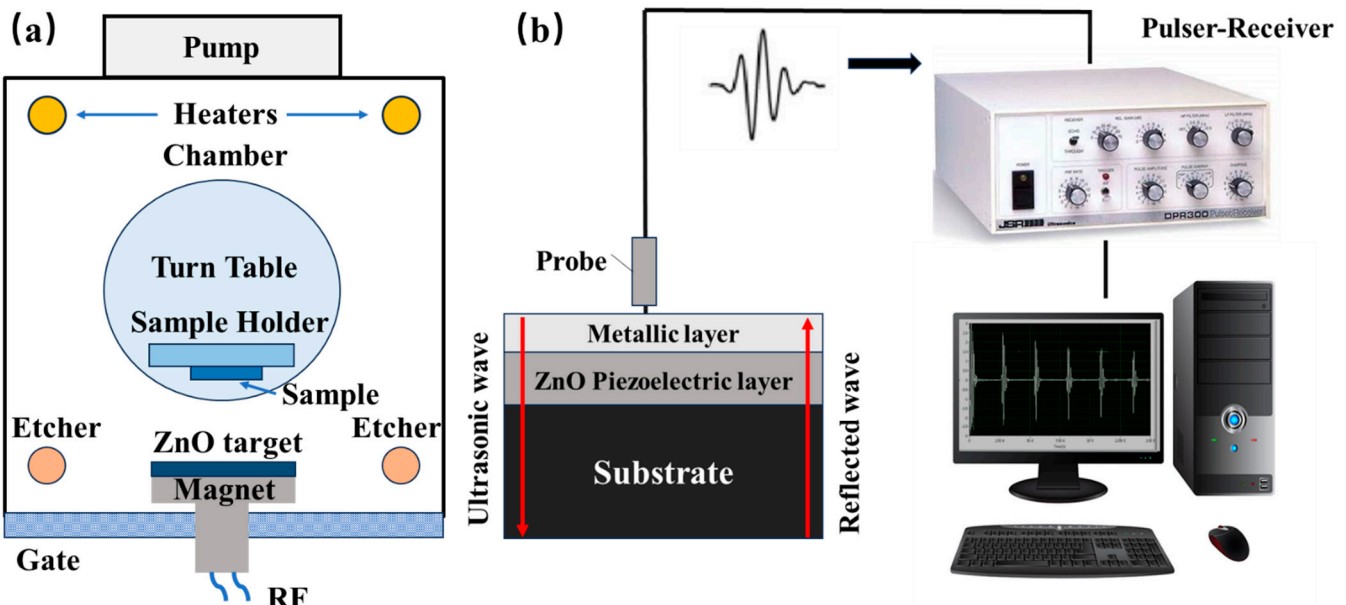

**Figure 1.** (**a**) Schematic diagram of the RF sputtering chamber; (**b**) schematic diagram of the experimental setup for ultrasonic response measurement.

### 2.2. Characterization

In this paper, the prepared ZnO piezoelectric thin-film materials were structurally characterized with an X-ray diffractometer, model Tongda TDM-10, using a ray source Cu Kα, λ = 0.15405 Å. In this paper, the surface and cross-section of the samples were morphologically characterized with a field emission scanning electron microscope, model MIRA3, and the thickness of the coatings was measured. The constituent elements of the samples were determined using an energy spectrometer (EDS). The hardness of the samples

was tested using an HVS-1000 A microhardness tester. The angle of its diamond relative surface was 136°, and the selected load was 500 g.

In this paper, the electrodes were prepared on the coating by spot-coating silver paste (purity 99.9%), and after the electrodes were dried and cured, an AC voltage was applied on both sides of the sample, and ultrasonic waves were received using a probe to realize the real-time measurement of the ultrasonic signals. A schematic diagram of the experimental setup for ultrasonic response measurement is shown in Figure 1b.

## 3. Results and Discussion

### 3.1. The Effect of Sputtering Powers on ZnO Piezoelectric Thin Films

3.1.1. Thin Films' Structure and Micromorphology

Figure 2 shows the surface and cross-section morphology of the ZnO piezoelectric films prepared under different sputtering powers. As can be seen from the figure, under the conditions of 100 W and 200 W, the surface of the prepared ZnO film showed uniformly arranged spherical particles. With the increase in sputtering power, the degree of ionization of argon ions increased, the sputtered particles had greater kinetic energy, the surface grain size increased, and the columnar crystals in the cross-section were also coarsened. By the time the sputtering power exceeded 200 W, the concentration of the sputtered particles increased. As the sputtering proceeded, the stable nuclei tended to merge with each other, which was due to the lattice distortion generating higher grain boundary energy, and clusters were formed between the stable nuclei of small sizes. As the sputtering power increased, the concentration of the particles increased, the grain size increased, and larger particles appeared on the surface of the thin film, which formed a denser crystalline structure, which can be seen in the cross-sectional morphology; the grain size of the columnar crystals was gradually coarsened, consistent with the morphology that extended to the surface.

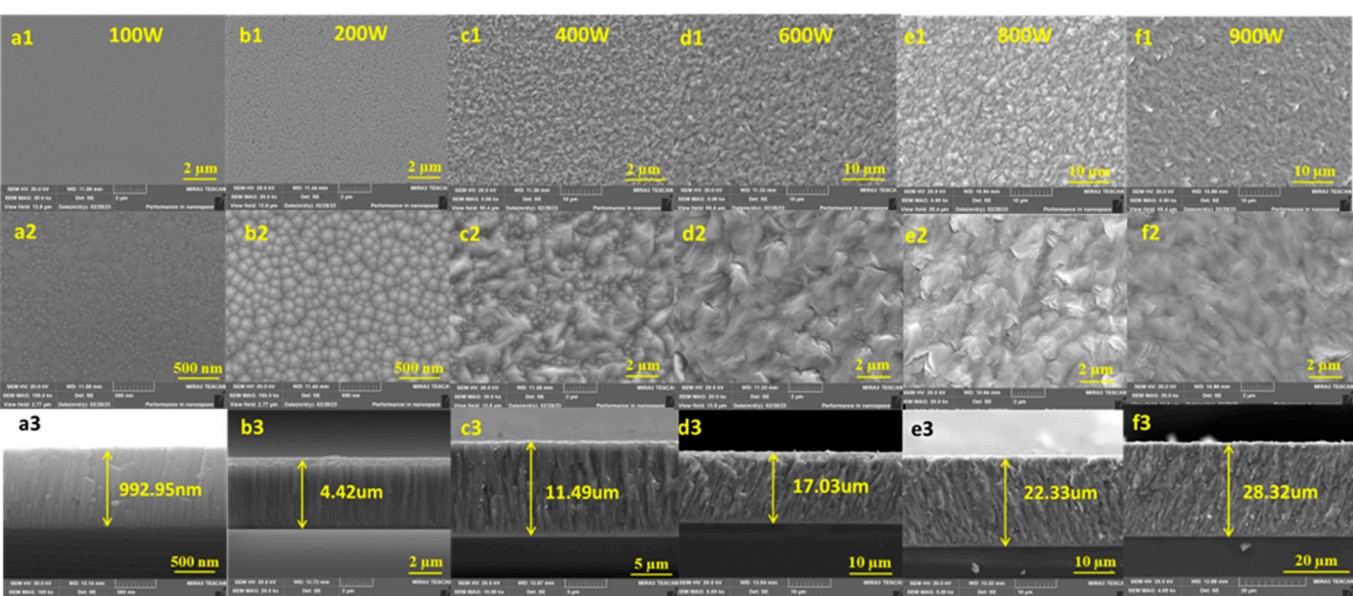

**Figure 2.** Surface and cross-section morphology of ZnO thin films under different sputtering powers: (**a1–a3**) 100 W; (**b1–b3**) 200 W; (**c1–c3**) 400 W; (**d1–d3**) 600 W; (**e1–e3**) 800 W; (**f1–f3**) 900 W.

Figure 3 shows the variation in the thickness and hardness of the ZnO thin films with different sputtering powers. As the sputtering power increased, the thickness of the ZnO film increased significantly, which was due to the fact that the increase in sputtering power caused the ionization of argon ions to increase, resulting in more atoms being sputtered out of the target and deposited onto the substrate. The increase in sputtering power also led to an increase in the average kinetic energy of the sputtered particles, and the violent

bombardment of the substrate by the particles resulted in a denser film. From the hardness change curve, it can be seen that when the sputtering was 100 W and 200 W, the hardness of the ZnO film stayed around 300 HV because it consisted of columnar crystals; when the sputtering power exceeded 200 W, the crystal structure of the ZnO film became denser, and the hardness of the ZnO film increased along with the increase in the power.

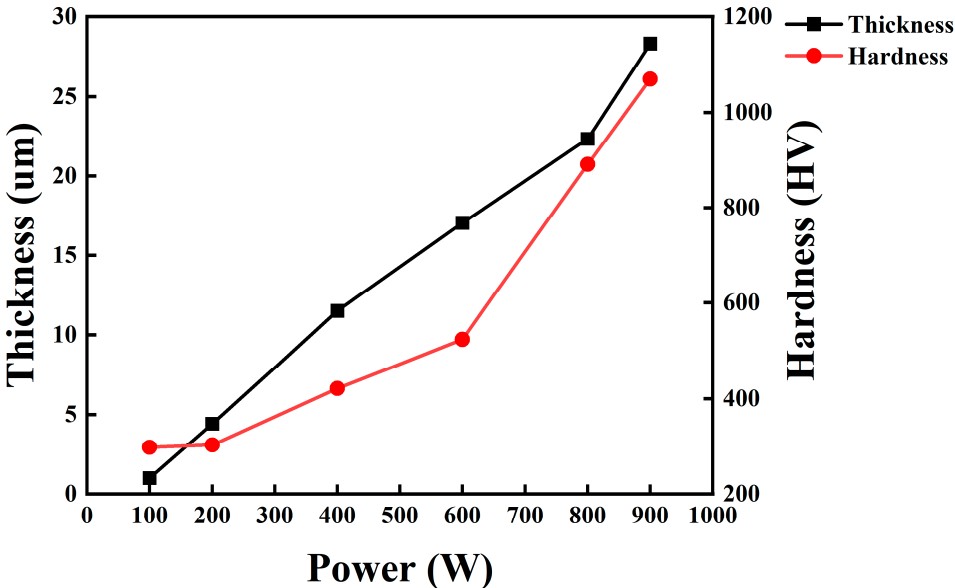

**Figure 3.** Variation in thickness and hardness of ZnO thin films with different sputtering powers.

Figure 4 shows the XRD patterns of the ZnO thin films on the stainless-steel substrate under different sputtering powers, as well as the peak intensity and full width at half maximum (FWHM) of its (002) diffraction peak. At lower sputtering power, the average kinetic energy of the sputtered particles was relatively small; also, the crystals could grow along the orientation with the lowest energy. Moreover, the ZnO films showed a tendency to grow in a single (002) orientation, which exhibited the growth of the high c-axis in a selective orientation. When the sputtering power exceeded 200 W, the excessively high energy caused the sputtered particles to move too fast, and the particles did not have enough time to migrate to the position with the lowest energy; in this case, diffraction peaks other than (002) peaks appeared in the XRD image, which showed a multiorientation growth structure. From the variation curves of peak intensity and half-peak width of the (002) peak with sputtering power, it can be seen that the peak intensity of the (002) peak was the largest and the half-peak width was the smallest when the sputtering power was 200 W, i.e., the content of crystalline phases was the highest in the ZnO thin films and the grain size was the largest in the (002) direction. In conclusion, the crystal growth of ZnO films is most favorable at a sputtering power of 200 W. When the power exceeds 200 W, the high kinetic energy of the sputtered particles affects the quality of film deposition.

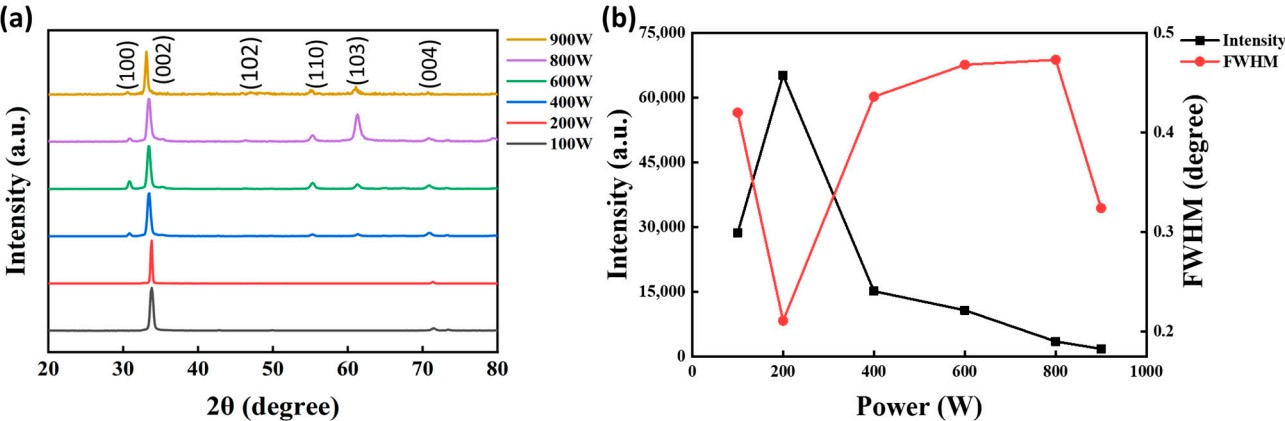

**Figure 4.** (**a**) XRD patterns of ZnO thin films grown at different sputtering powers; (**b**) intensity and FWHM of (002) diffraction peak at different sputtering powers.

### 3.1.2. Ultrasonic Signal Performance

The speed of propagation of ultrasound is determined by the medium of propagation. Ultrasonic longitudinal waves (LWs) in stainless steel have a propagation speed of 5880–5950 m/s, and ultrasonic transverse waves (SWs) in stainless steel have a propagation speed of 2940–2990 m/s; the type of excited ultrasonic waves and their different propagation speeds can be judged by calculating the time of flight from the figure [14]. Figure 5 shows an image of ultrasonic signals excited by the ZnO piezoelectric films prepared under the power of 400 W to 900 W. The samples under the power conditions of 100 W and 200 W failed to excite ultrasonic signals under the AC voltage, which was due to the fact that the ZnO piezoelectric films under these conditions were thin and the deformation amplitude induced by the AC voltage was too small for the desired ultrasonic signals to be excited. When the thickness of the film reached 10 um and above, the intensity of the excited ultrasonic signal was more ideal. The ZnO piezoelectric films prepared in the power range of 400 W to 900 W were able to excite ultrasonic signals with mixed longitudinal and transverse waves. This was due to the fact that when the sputtering power is greater than 200 W, ZnO films show multiorientation growth behavior, which provides conditions for the generation of transverse waves [26,27].

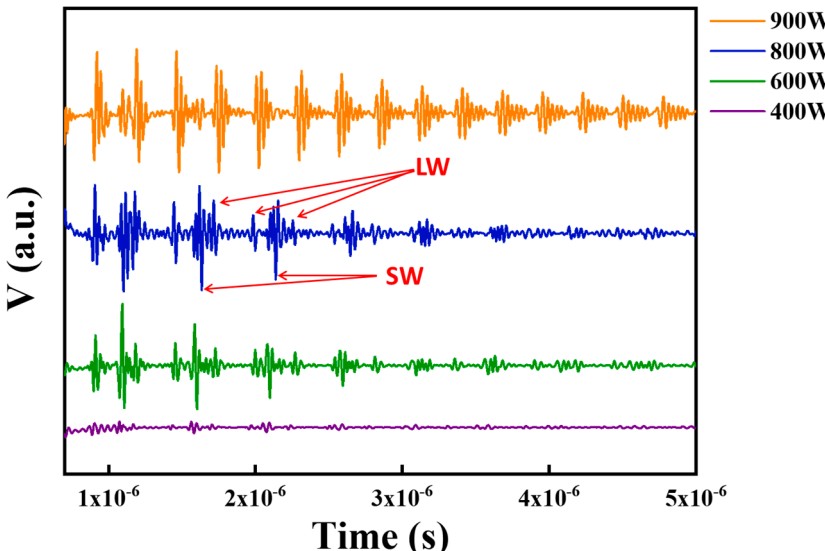

**Figure 5.** The ultrasonic signal of ZnO thin films with different sputtering powers.

Figure 6a shows the waveform and frequency spectrum of the longitudinal wave signals of the ZnO films under different sputtering powers; Figure 6b shows the variation in

the amplitude and frequency peaks of the longitudinal wave signals of the ZnO piezoelectric films with the sputtering power. There was a certain positive correlation between the amplitude of the ultrasonic signals that could be excited by the ZnO piezoelectric films and the thickness of the films. With the increase in sputtering power, the thickness of the films increased, so the intensity of the longitudinal wave signals became stronger. From the spectral characteristics curve, it can be seen that under different film thicknesses, the ZnO piezoelectric film was able to excite ultrasonic longitudinal signals around 30 MHz, which is sufficient to meet the needs of measuring bolt preload.

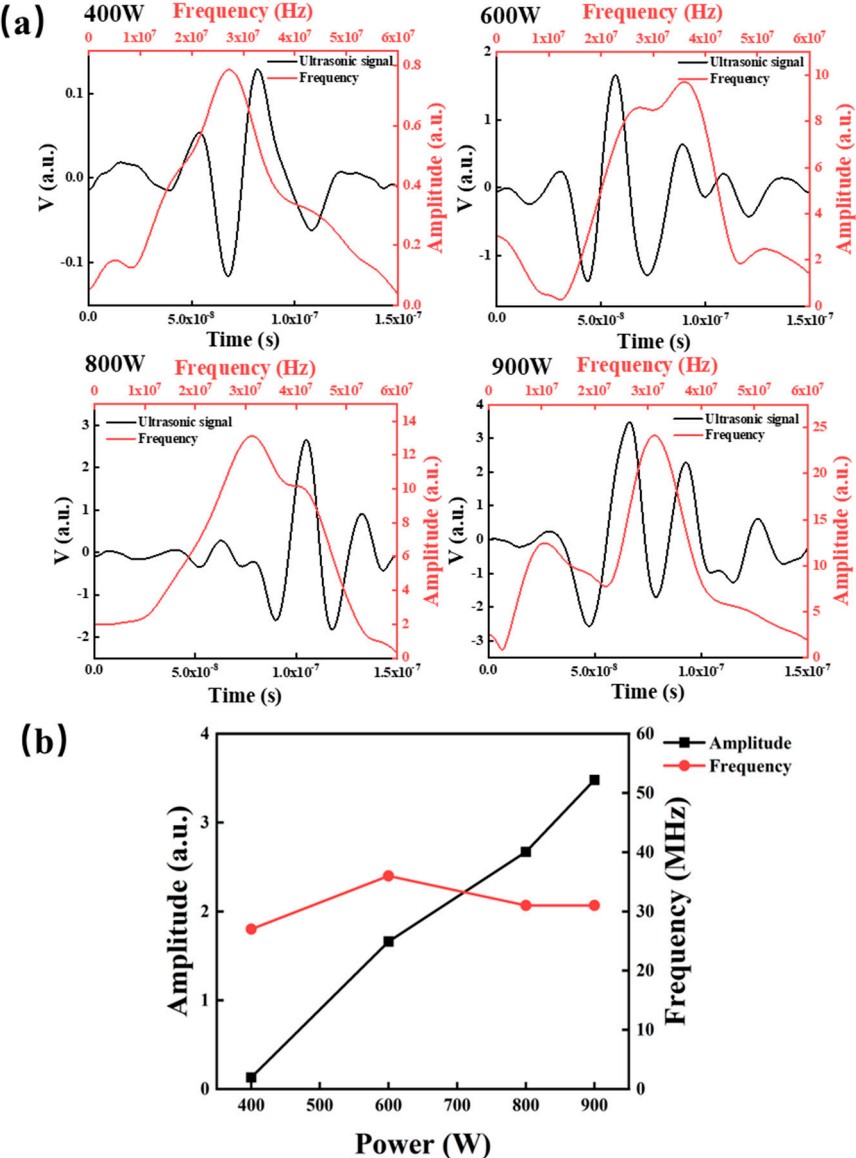

**Figure 6.** Longitudinal wave signals of ZnO thin films under different sputtering powers. (**a**) Waveform and frequency spectrum; (**b**) amplitude and frequency.

### 3.2. The Effect of Target Substrate Distances on ZnO Piezoelectric Thin Films

### 3.2.1. Thin Films' Structure and Micromorphology

When the sputtering power exceeded 200 W, the growth of the ZnO thin films showed a multiorientation growth structure and failed to reach a single (002) orientation growth state due to the high kinetic energy of the sputtered particles. In addition to reducing the sputtering power, increasing the distance between the target and the substrate could also achieve the effect of reducing the kinetic energy of the sputtered particles. Figure 7

shows the surface and cross-section morphology of the ZnO thin-film samples prepared at different target substrate distances. Under the condition of 600 W sputtering power, when the target spacing was increased to 80 mm and 100 mm, the surface morphology of the ZnO film had a tightly arranged spherical morphology, which corresponded to the morphology of the cross-section, shown as a tightly arranged columnar crystalline structure with clear grain boundaries.

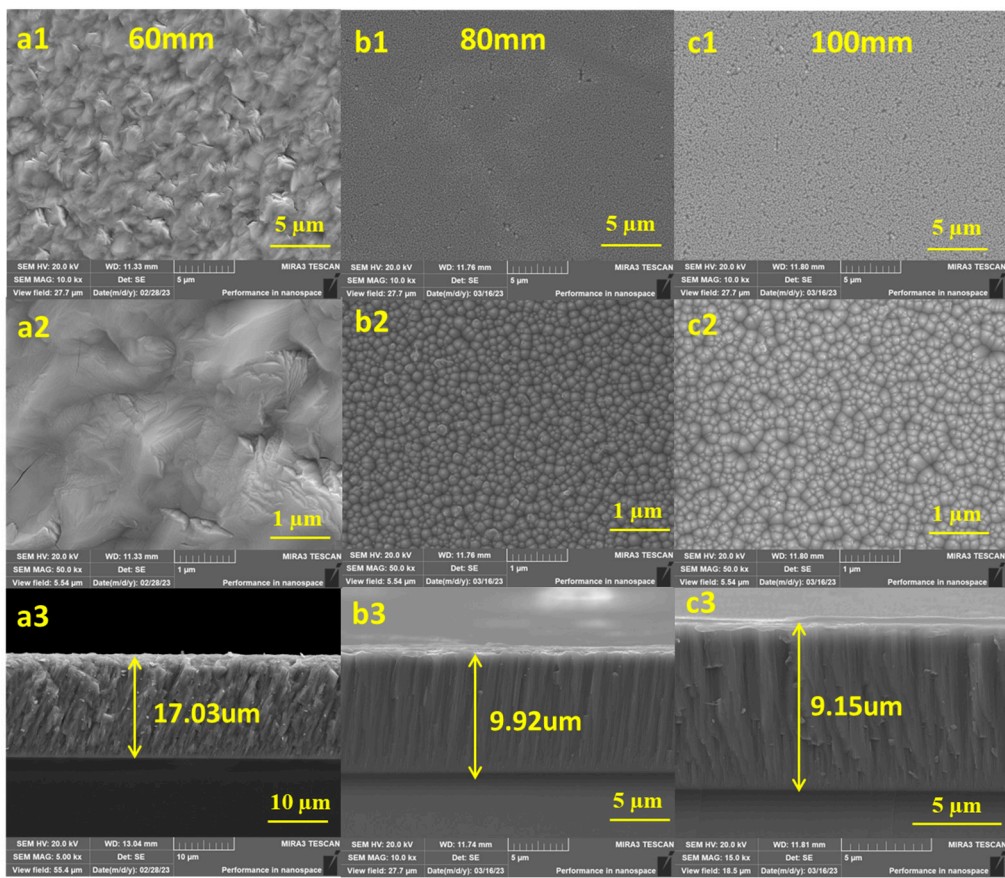

**Figure 7.** Surface and cross-section morphology of ZnO thin films under different target substrate distances: (**a1**–**a3**) 60 mm; (**b1**–**b3**) 80 mm; (**c1**–**c3**) 100 mm.

Figure 8 shows the variation in thickness and hardness of the ZnO films with different target substrate distances. As the distance between the target and the substrate increased, the particles were subjected to more collisions before reaching the substrate, so the number of particles reaching the substrate decreased with the increase in the target substrate distance; thus, the deposition rate of the ZnO thin film decreased, i.e., the thickness of the sample with a large target substrate distance was smaller. At the same time, with the increase in the target substrate distance, the intensity of sputtering particles bombarding the substrate was weakened, the density of the deposited ZnO film decreased, and the hardness of the film decreased with the increase in the target substrate distance.

Figure 9 shows the XRD patterns of ZnO thin films under different target substrate distance conditions, as well as the peak intensity and FWHM of the (002) diffraction peak. From the figure, it can be seen that the growth trend of ZnO films changed from multiorientation growth dominated by a (002) orientation to single (002) orientation growth as the target substrate distances increased. The changes in peak intensity and FWHM indicate that, with the increase in the target substrate distance, the peak intensity of the (002) diffraction peak increased, and the content of crystalline phases in the structure of the ZnO thin films increased, the half-peak width decreased, and the average grain size of the samples increased, which significantly improved the crystalline quality of the thin films.

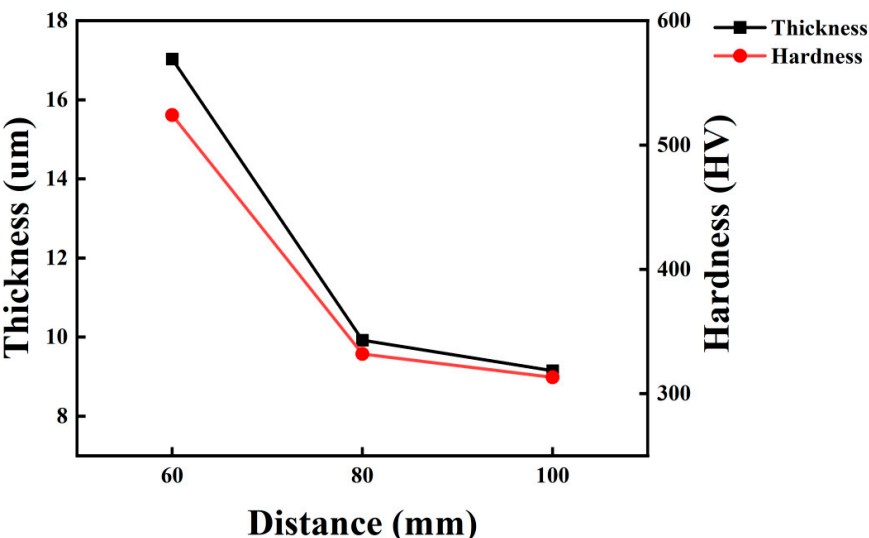

**Figure 8.** Variation in thickness and hardness of ZnO films with different target substrate distances.

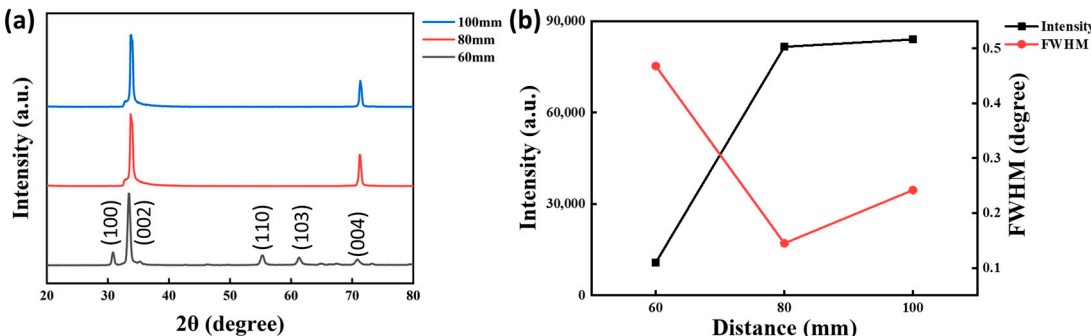

**Figure 9.** (**a**) XRD patterns of ZnO thin films grown at different target substrate distances; (**b**) intensity and FWHM of (002) diffraction peak at different target substrate distances.

### 3.2.2. Ultrasonic Signal Performance

Figure 10 shows the ultrasonic signals of ZnO piezoelectric films at different target substrate distances. At a target substrate distance of 60 mm, the type of ultrasonic signal excited by the sample was a mixture of longitudinal and transverse waves, which was due to the structure of the ZnO film; for a multiorientation growth structure, it can be easy to stimulate ultrasonic longitudinal and transverse mixed waves. After increasing the target substrate distance to 80 mm, the sample stimulated more ideal ultrasonic longitudinal signals; this was related to the conditions of the film's single (002) orientation of growth [28,29]. We continued to increase the target substrate distance to 100 mm. When the target substrate distance was increased to 100 mm, the excitation signals of the samples were mainly longitudinal ultrasonic waves, and the intensity of the ultrasonic signals of the samples under the 100 mm target substrate distance condition was weaker than that under the target substrate distance of 80 mm due to the reduction in the thickness of the samples.

Figure 11a shows the waveform and frequency spectrum of the longitudinal wave signal of the ZnO thin films with different target substrate distances, and the variation in the amplitude and frequency peaks of the longitudinal wave signal with the target substrate distance is shown in Figure 11b. As the target pitch increased, the number of particles arriving at the substrate during the sputtering process decreased, the thickness of the ZnO film deposited decreased in the same period of time, and the intensity of the ultrasonic signals that could be excited by the samples was weakened. The ZnO piezoelectric films prepared under the 100 mm target substrate distance condition had better frequency characteristics and could excite an ultrasonic longitudinal signal of 41 MHz. By increasing

the target substrate distance, the average kinetic energy of the sputtered particles was reduced, which could achieve the effect of controlling the growth of the ZnO piezoelectric films, and then regulated the ultrasonic performance of the ZnO piezoelectric films. In conclusion, under the conditions of 600 W sputtering power and a target substrate distance of 100 mm, a ZnO piezoelectric thin-film material with better ultrasonic performance can be produced.

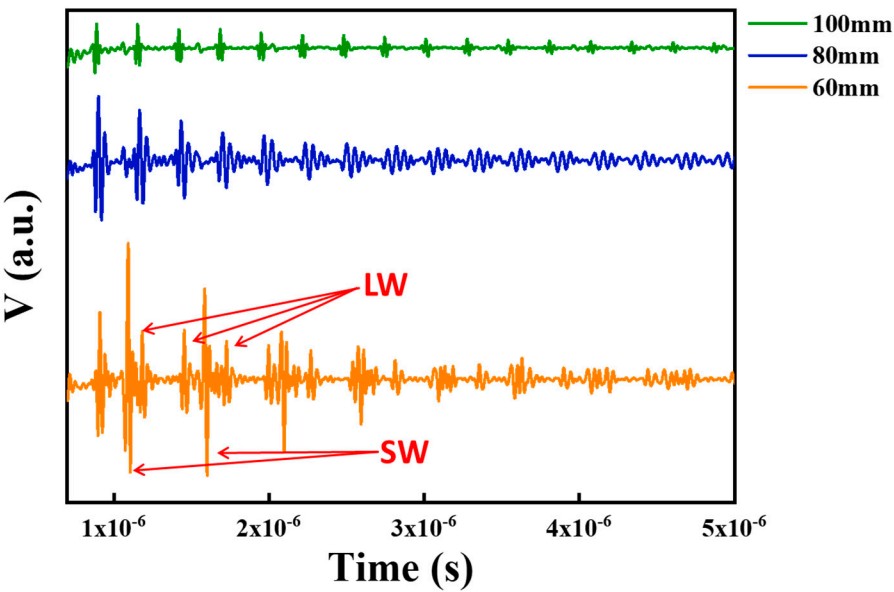

**Figure 10.** The ultrasonic signal of ZnO thin films with different target substrate distances.

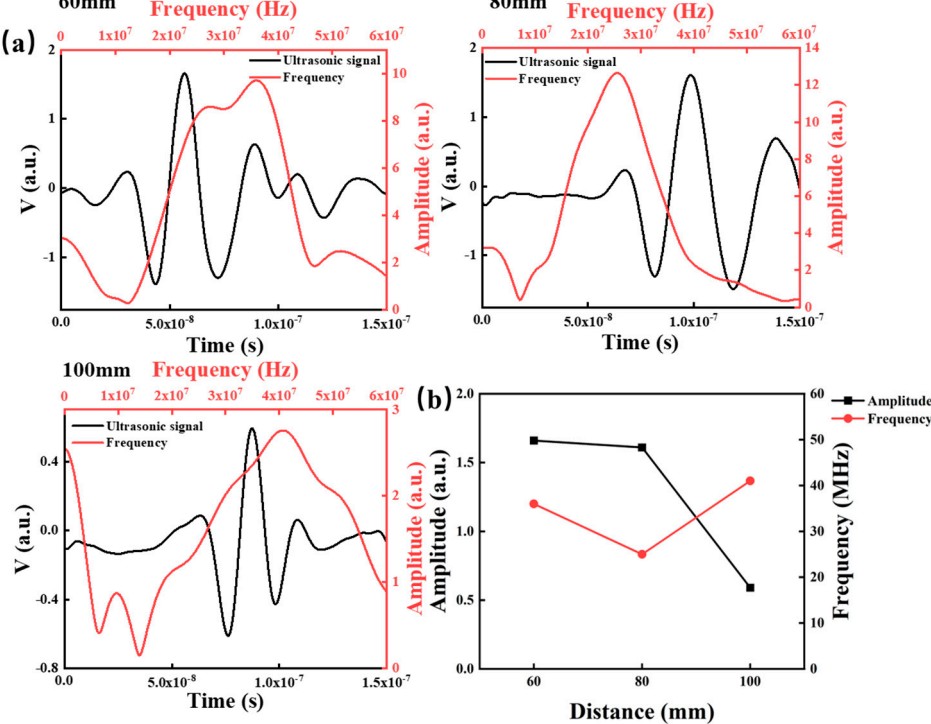

**Figure 11.** Longitudinal wave signals of ZnO thin films under different target substrate distances. (**a**) Waveform and frequency spectrum; (**b**) amplitude and frequency.

## 4. Conclusions

In this paper, ZnO piezoelectric thin-film materials were prepared using RF magnetron sputtering under different process parameters; also, the effects of sputtering power and target substrate distance on the structure and properties of the ZnO piezoelectric thin films were tested. The experimental results show that too-large sputtering power or a too-small target substrate distance leads to excessive kinetic energy in particles in the sputtering process, and a too-fast deposition rate prevents crystals from growing according to the lowest (002) orientation; the crystals show a tendency to grow in multiple orientations. The high kinetic energy of the particles leads to the formation of a dense crystal structure during deposition, while the multioriented crystal growth and dense crystal structure are not favorable to the generation and propagation of pure ultrasonic longitudinal waves. A ZnO piezoelectric film prepared under the parameters of 600 W sputtering power, 4 h sputtering time, and 100 mm target substrate distance is able to stimulate ultrasonic longitudinal wave signals with the ideal intensity under AC voltage; its operating frequency is 41 MHZ, which meets the requirements of the technology of using ultrasonic longitudinal waves to measure the preload force of bolts. Therefore, by adjusting the preparation process of ZnO piezoelectric thin films, the growth orientation of the films can be directly affected, thus adjusting the ultrasonic excitation performance of the film.

**Author Contributions:** Conceptualization, Y.J.; Methodology, Y.J.; Software, C.M.; Investigation, Y.Z.; Resources, C.M.; Writing—original draft, Y.Z. and Y.J.; Writing—review & editing, Y.J., J.Z. and B.Y.; Project administration, J.Z.; Funding acquisition, B.Y. All authors have read and agreed to the published version of the manuscript.

**Funding:** This research was funded by Seed Fund Program for Sino-Foreign Joint Scientific Research Platform of Wuhan University: [Grant Number WHUZZJJ202223]; Science and Technology Planning Project of Shenzhen Municipality: [Grant Number 2022-047].

**Institutional Review Board Statement:** Not applicable.

**Informed Consent Statement:** Not applicable.

**Data Availability Statement:** Not applicable.

**Conflicts of Interest:** The authors declare no conflict of interest.

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
