# Peer review of "Preparation of ZnO Piezoelectric Thin-Film Material for Ultrasonic Transducers Applied in Bolt Stress Measurement"

_coatings, doi:10.3390/coatings13091538_

Round 1

Reviewer 1 Report

The authors present interesting work on ZnO piezoelectric thin films for application as ultrasonic transducers in bolt stress measurements. The article is well-written and will be a good reference for researchers working on the topic. Reviewer recommends the publication of the article with the following suggestions.

1.      In the introduction, please highlight the advantages of the ZnO thin film as an ultrasonic transducer for the specified application compared with other thin film piezo materials, e.g., GaN or PZT.

2.      Article is about the bolt stress measurements; please add the schematic of the bolt test measurement along with the actual ZnO piezo transducer. Please add the stress measurement calculations and corresponding formulations for it.

3.      Why is the ZnO film's ultrasonic signal shown in the V arbitrary unit? The actual value of the measurement should be shown and discussed.

4.      Why did the author choose Si and metal substrates? Schematic of the device and connection should be discussed in the manuscript.

5.      Section 3.2 why the 600 W power was selected for the variation of target to substrate distance, the obtained morphology looked like the 200W power sample. So if a 200W power processed sample, if sputtered for a longer time, could have produced the thicker films, it might produce similar results.

6.      w.r.t figures

o   Please use readable, high-resolution images, for eg. For Fig 6 a, Fig. 11 a

o   Fig, 4 and 9, would be better if the axis are zoomed in around important peaks eg. 002 30 -40 d in the main article.

o   Fig 5 is repeated. Define all abbreviations eg. SW, LW in Fig. 5, Fig. 10

o   Please check the label eg. Fig 7 (Fig 2)

7.      Please check spelling (eg. arget in fig. 9 caption) and sentence formations (for e.g. Line 49 “the piezoelectric film of the electromechanical coupling method”)

8.      For broader readership topics relevant to ZnO growth by a sputtering method for energy sensing devices are suggested to refer and cite, e.g., ZnO-based transparent optoelectronics in https://doi.org/10.1016/j.solmat.2019.02.004, defect-mediated ZnO film for photovoltaics in https://doi.org/10.1016/j.apmt.2021.101344 , ZnO active energy window for building integrated device in https://doi.org/10.1016/j.xcrp.2021.100591.

Please check spellings and sentence formations

Reviewer 2 Report

This paper aims to investigate the fabrication process of ZnO piezoelectric film, its geometric structures, and their corresponding ultrasonic wave characteristics. The findings of this study hold significant interest and merit consideration for publication in the Coating journal.

(1) In the introduction section, it is important to elucidate the rationale behind selecting ZnO as the preferred material for the piezoelectric film over alternatives such as PZT and BaTiO3. Although PZT and BaTiO3 exhibit remarkable piezoelectric properties and demonstrate favorable temperature insensitivity, a comprehensive explanation is necessary for the preference of ZnO in this specific study.

(2) With regards to Figure 6, it is essential that the correlation between the morphology and structure of ZnO and the resultant ultrasonic characteristics (amplitude and frequency) be established, rather than focusing on the source power. Similarly, in Figure 10, the discussion should revolve around the direct relationship between the geometric structure of the ZnO film and the generation of ultrasonic waves, as opposed to attributing it to power source or the distance between the target and substrate. Consequently, a detailed elaboration on the mechanism underlying the generation of ultrasonic waves based on the film's geometry should be prominently considered in the discussion
